# Associations between job demands, job resources and patient-related burnout among physicians: results from a multicentre observational study

Renée Scheepers  ,[1,2] Milou Silkens,[3] Joost van den Berg,[4] Kiki Lombarts[2]

[1]Erasmus School of Health Policy and Management, Erasmus University Rotterdam, Rotterdam, The Netherlands
[2]Medical Psychology, Amsterdam University Medical Centres, Amsterdam, Noord-Holland, The Netherlands
[3]Research Department of Medical Education, University College London, London, UK
[4]Internal Medicine, Amsterdam University Medical Centres, Amsterdam, Noord-Holland, The Netherlands

**Correspondence to**
Dr Renée Scheepers;
scheepers@eshpm.eur.nl

## ABSTRACT

**Objectives** To investigate associations of job demands and resources with patient-related burnout among physicians.

**Design** Multicentre observational study.

**Setting** Fifty medical departments at 14 (academic and non-academic) hospitals in the Netherlands.

**Participants** Four hundred sixty-five physicians (71.6% response rate), comprising 385 (82.8%) medical specialists and 80 (17.2%) residents.

**Main outcome measures** Job demands (workload and bureaucratic demands), job resources (participation in decision making, development opportunities, leader's inspiration, relationships with colleagues and patients)—measured with the validated Questionnaire of Experience and Evaluation of Work and Physician Worklife Survey—and patient-related burnout, measured using the validated Copenhagen Burnout Inventory.

**Results** Patient-related burnout was positively associated with workload (b=0.36; 95% CI, 0.25 to 0.48; p<0.001) and negatively associated with development opportunities (b=−0.18; 95% CI, −0.27 to −0.08; p<0.001) and relationships with patients (b=−0.12; 95% CI, −0.22 to −0.03; p=0.01). Relationships with patients moderated the association between bureaucratic demands and patient-related burnout (b=−0.15; 95% CI, −0.27 to −0.04; p=0.01).

**Conclusions** Physicians with high workloads and few development opportunities reported higher levels of patient-related burnout. Those with positive patient relationships were less likely to experience patient-related burnout, even in the presence of excessive bureaucracy. Therefore, positive physician–patient relationships may be supported to reduce the likelihood of physicians' patient-related burnout. However, the specific support needed to effectively reduce patient-related burnout may vary per healthcare context and thus requires intensified research across healthcare systems and settings.

## INTRODUCTION

In the last decade, research has revealed risks of physicians' burnout to patient care quality.[1 2] Patient care quality is more likely to be suboptimal—as evidenced by increased numbers of safety incidents and lower levels of patient satisfaction—when physicians

---

**Strengths and limitations of this study**

► This study addressed a knowledge gap on how job demands and job resources are associated with patient-related burnout among physicians.

► Job demands, job resources and patient-related burnout were measured by validated instruments that were selected based on both a needs assessment among practicing physicians and the evidence-based job demands and resources model.

► This multicentre study was conducted in academic and non-academic medical centres and included multiple specialties, to warrant generalisability of findings to diverse hospital-based settings.

► Our study resulted in a substantial response rate (71.6%) and accounted for potential confounders, though our findings may have been affected by self-selection of participants.

► The cross-sectional study design precluded the assessment of causal associations, yet our cross-sectional findings did align with longitudinal findings of related research.

---

are burned out.[1 2] Physicians' burnout rates are high worldwide, and this situation has been recognised as a global system-level problem.[3–7] Across healthcare systems, physicians' burnout has been related to the stressful working conditions of modern medical practice, which involve heavy workloads in combination with constant time pressure and an excessive administrative burden.[8–10] These conditions sap physicians' energy by reducing autonomy and by limiting physicians' time for and attention to patients.[11 12] Connecting with patients—the very essence of being a physician—has become increasingly challenging in modern practice.

This connection with patients and the provision of patient care used to be the main source of physicians' professional satisfaction and sense of meaning in work.[12] Recently, however, physicians have reported exhaustion in providing patient care.[13 14] The degree of

exhaustion that physicians relate to working with patients indicates patient-related burnout, while work-related burnout involves the exhaustion that professionals attribute to their work in general.[15] Physicians generally report higher levels of work-related than patient-related burnout.[14 16 17] However, even low levels of patient-related burnout can be problematic as they indicate physicians' exhaustion in their core task—caring for patients. Furthermore, patient-related burnout has been associated with absenteeism due to sickness.[18] Physicians report higher levels of patient-related burnout when exposed to work environments with higher quantitative demands.[14 19] In general, burnout is more likely to develop when job demands—stressful aspects of work (eg, workloads)—are high. On the other hand, burnout may be minimised or prevented by the provision of job resources—energising aspects of work (eg, development opportunities).[20–22]

Insight into which job demands and resources are related to physicians' patient-related burnout in particular, however, remains limited; few studies on this topic have been conducted to date.[14 15] These studies have not involved the examination of job demands and resources that are specifically relevant to medical practice (eg, excessive bureaucracy and relationships with patients). Moreover, these studies have not yet clarified how job demands and resources interact in relation to patient-related burnout; this interaction may matter in the context of burnout prevention as job demands less likely result in burnout when job resources are high.[20] To aid the targeting of relevant components, the objective of the current study was to investigate associations of job demands and resources to physicians' patient-related burnout, and to clarify the interaction of demands and resources in this context.

## METHODS
### Study population and setting
This study investigated associations between job demands, job resources and patient-related burnout by conducting a nationwide programme involving a survey on perceived working conditions and well-being of medical staff in 50 departments at 14 hospitals in the Netherlands from April 2017 to June 2018. With an email describing the programme, we invited 649 physicians to complete an online survey. Participation was voluntary, and participants' anonymity and confidentiality were safeguarded.

### Patient and public involvement
There were no patients involved in the study. Physicians were involved in the design of the study based on the needs assessment described below. The findings have been disseminated through oral presentations at conferences for physicians and through newsletters on the website of the Professional Performance and Compassionate Care Research Group (https://professionalperformance-amsterdam.com/en/). This platform also made the measurement tools on working conditions and well-being publicly available to hospitals.

### Measures
The survey included validated questionnaires about job demands, job resources and patient-related burnout. As job demands and resources vary across professional settings, we selected those most relevant to physicians' practice. This selection was based on the validated job demands and resources model[21 23 24] and on a needs assessment, consisting of two focus groups and a web-based survey. The two focus groups included 24 participants in total (physicians and residents) and explored potentially relevant job resources and demands; the web-based survey was completed by 218 participants (physicians and residents), who assigned priority to the most relevant job demands and resources in medical practice. The results of the focus groups and survey were discussed in the research team, leading to the ultimate selection of job demands and resources (see below).

Job demands included workload and bureaucratic demands. Workload was measured using the validated 6-item subscale on workload of the Questionnaire on the Experience and Evaluation of Work (QEEW), with responses structured by a 4-point scale ranging from 1 ('never') to 4 ('always').[25] Bureaucratic demands were measured by the validated 3-Item Red Tape Scale, consisting of one item ("How would you describe policies and procedures in your work division between the following opposite characteristics?'), with responses given on 5-point scales ranging from 'not burdensome' to 'burdensome', 'necessary' to 'unnecessary', and 'effective' to 'ineffective'.[26]

Job resources included participation in decision making, development opportunities, leaders' inspiration, relationships with colleagues and relationships with patients. The first four resources were measured using the QEEW[25] and the fifth was measured using the validated Physician Worklife Survey.[27] Responses to the three items on relationships with colleagues and four items on the leader's inspiration were structured by a 4-point scale ranging from 1 ('never') to 4 ('always'). Those to the four items each on participation in decision making, development opportunities and relationships with patients were given on a 5-point scale ranging from 1 ('totally disagree') to 5 ('totally agree'). As the original relationships with patients subscale had not been validated in Dutch, two researchers independently translated the English version into Dutch and agreed on the Dutch version, which another bilingual researcher back-translated. We resolved minor differences between the back translation and original, adjusting the forward translation to create the final Dutch version of the subscale.

Patient-related burnout was measured using the validated 6-item Copenhagen Burnout Inventory, with responses structured by a 5-point scale ranging from 1 ('totally disagree') to 5 ('totally agree').[15]

The survey also included questions about physicians' characteristics: training level (specialist/resident), years of experience, specialty (surgical/medical), type of employment (full-time/part-time) and sex (male/female). These data were included in the statistical analysis to adjust for potential confounding of associations of job resources and demands with patient-related burnout.

## Statistical analysis

Sample characteristics were represented using frequencies and descriptive statistics. The psychometric properties of the job demands, job resources and patient-related burnout constructs were assessed using exploratory factor analysis (EFA) and reliability analysis (online supplemental tables A1 and A2). For the EFA, we performed principal axis factoring with oblique rotation and chose the Kaiser-Guttman criterion and fixed factor models for extraction of the optimal number of factors.[28] The research team discussed the theoretical relevance of two job resources items with loadings <0.40, and decided to retain them because they originated from validated questionnaires[25 27] and were considered to contribute meaningfully to the overall construct. The EFA yielded two job demands subscales (9 items), five job resources subscales (20 items) and one patient-related burnout subscale (6 items; online supplemental table A1). Reliability was assessed according to internal consistency (satisfactory when Cronbach's $\alpha > 0.70$,[29] interscale correlations (satisfactory when Pearson's $r < 0.70$) and the item-total correlations (satisfactory when Pearson's $r > 0.30$). Following the establishment of construct validity and reliability, mean subscale and total scores were calculated for each construct.

To assess the associations of job demands and resources with physicians' patient-related burnout, we used unadjusted and adjusted random-intercept generalised linear mixed models. These models allowed us to account for the hierarchical clustering of individuals within clinical departments within hospitals. In the unadjusted models, total mean scores (model 1) and individual subscale scores (model 2) for the job demands and resources constructs served as independent variables, and scores on patient-related burnout served as the dependent variable. Subsequent models were adjusted for physicians' characteristics, that is, physicians' sex, post–Doctor of Medicine degree years of experience, employment type (full-time/part-time) and type of respondent (medical specialist/resident).[30]

Furthermore, we analysed interactions between job demands and resources in relation to patient-related burnout by employing moderation analysis, using the SPSS macro PROCESS.[30] Specifically, we conducted multiple regression analyses including the independent variables (1) a specific job demand (workload or bureaucratic demands), (2) a specific job resource (participation in decision making, development opportunities, leaders' inspiration, relationships with colleagues or relationships with patients), (3) the interaction term of

the respective job demand and resource, and (4) patient-related burnout as the dependent variable. A significant interaction term indicated a moderation effect, which was inspected by performing simple slopes analysis to measure the conditional effects of the independent variable for three values of the moderator: (1) low score (−1 SD), (2) the average score and (3) high score (+1 SD). All results were reported using regression coefficients (b), their 95% CI and p values (<0.05). All analyses were performed using SPSS Statistics V.25.

## RESULTS

In total, 465 physicians (82.8% specialists, 17.2% residents) from 50 clinical departments at 16 hospitals completed the questionnaire (71.6% response rate; table 1). Of them, 111 (23.9%) physicians originated from academic hospitals.

The job demands, job resources and patient-related burnout subscales showed satisfactory to good internal consistency, and interscale as well as item-to-total correlations (online supplementary tables A1 and A2). Our analyses of associations between job demands, job resources and patient-related burnout showed that the unadjusted model 1 revealed significant associations of job demands and resources with patient-related burnout, confirmed by the adjusted model 2 (table 2). The unadjusted model

**Table 1** Characteristics of the study population

| Characteristics | N (%) |
|---|---|
| Number of respondents | 465 (100) |
| Male | 222 (47.7) |
| Female | 243 (52.3) |
| Type of respondent | |
| Medical specialist | 385 (82.8) |
| Resident | 80 (17.2) |
| Specialty | |
| Surgical | 193 (41.5) |
| Non-surgical | 226 (48.6) |
| Supporting | 32 (6.9) |
| Non-medical | 14 (3.0) |
| Years after completing MD | |
| 0–5 | 40 (8.6) |
| 6–10 | 93 (20.0) |
| 11–15 | 88 (18.9) |
| 16–21 | 80 (17.2) |
| 22–45 | 161 (34.6) |
| 46+ | 3 (0.6) |
| Type of contract | |
| Full-time | 257 (55.3) |
| Part-time | 208 (44.7) |

MD, Doctor of Medicine.

**Table 2** Unadjusted and adjusted models predicting the effect of job resources and job demands on patient-related burnout by using total mean scores (model 1) or subscale scores (model 2)

| | Unadjusted model | Adjusted model* |
|---|---|---|
| | Regression coefficient (95% CI; p value) | Regression coefficient (95% CI; p value) |
| Job resources | | |
| Total mean score | −1.15 (−0.28 to −0.01; 0.03) | −0.17 (−0.31 to −0.04; 0.01) |
| Relationships with colleagues | 0.09 (−0.04 to 0.22; 0.17) | 0.08 (−0.06 to 0.21; 0.26) |
| Participation in decision making | 0.02 (−0.07 to 0.11; 0.73) | 0.02 (−0.08 to 0.11; 0.72) |
| Development opportunities | −0.17 (−0.26 to −0.07; 0.00) | −0.18 (−0.27 to −0.08; 0.00) |
| Leaders' inspiration | 0.04 (−0.04 to 0.12; 0.30) | 0.03 (−0.05 to 0.11; 0.41) |
| Relationships with patients | −0.12 (−0.22 to −0.03; 0.01) | −0.12 (−0.22 to −0.03; 0.01) |
| Job demands | | |
| Total mean score | 0.29 (0.17 to 0.42; 0.00) | 0.29 (0.17 to 0.42; 0.00) |
| Workload | 0.34 (0.23 to 0.45; 0.00) | 0.36 (0.25 to 0.48; 0.00) |
| Bureaucratic demands | 0.09 (0.01 to 0.17; 0.03) | 0.08 (−0.00 to 0.16; 0.06) |

*Adjusted for: physicians' sex, years of experience after obtaining an MD, type of contract (full-time/part-time) and type of respondent (medical specialist/resident).
MD, Doctor of Medicine.

2 showed that patient-related burnout was associated significantly with the two job demands subscales of workloads and bureaucracy and the job resources subscales of development opportunities and relationships with patients. The job resources subscales of relationships with colleagues, participation in decision making and leaders' inspiration were not associated with patient-related burnout (table 2). All of these associations except that with the job demand subscale of bureaucracy were confirmed by the adjusted model (table 2).

Relationships with patients significantly moderated the relationship between bureaucratic demands and patient-related burnout ($b_{interactionterm}$=−0.15; 95% CI, −0.27 to −0.04; p=0.01). Specifically, bureaucratic demands were significantly positively associated with patient-related burnout when physicians reported low (−1 SD; mean of 3.25) or average (mean of 4.00) ratings on the quality of their relationships with patients. Specifically, the association between bureaucratic demands and patient-related burnout was stronger when physicians' ratings of relationships with patients were low ($b_{low}$=0.21; 95% CI, 0.11 to 0.31; t=4.18) than when ratings were average (p<0.001 and $b_{average}$=0.10; 95% CI, 0.02 to 0.18; t=2.35; p=0.02). When physicians' ratings of patient relationships were high, there was no association between bureaucratic demands and patient-related burnout ($b_{high}$=0.02; 95% CI, −0.09 to 0.13; t=0.34; p=0.73).

## DISCUSSION
### Principal findings
This study on associations between job demands, job resources and patient-related burnout showed that physicians with high workloads and few development opportunities report higher levels of patient-related burnout.

Those with positive patient relationships are less likely to experience such burnout, even in the presence of excessive bureaucracy.

### Strengths and limitations of the study
This multicentre study used widely validated instruments to examine associations between job demands, job resources and patient-related burnout, which were selected based on needs assessment of practicing physicians' needs in alignment with the theoretical assumptions of the job demands and resources model. However, other job demands and resources also may be relevant, especially in other settings (eg, non-Dutch systems and primary care). Furthermore, our findings may not be generalisable to other healthcare systems, although our findings align with previous findings of related research—on associations between job demands, resources and burnout—in diverse healthcare systems.[22 24 31–33] Nonetheless, intensified research on the role of physicians' characteristics—across diverse systems and settings—is needed to clarify whether and how findings on job demands, resources and patient-related burnout should be tailored to specific physician subgroups.

In our study, the study sample was characterised by a high response rate, inclusion of physicians from multiple specialties and the sex distribution of the sample was consistent with national data.[34] As study participation was voluntary, subscale scores may be subject to self-selection bias. However, the observed associations of job demands and resources with patient-related burnout—the main focus of the current study—are in agreement with related findings from diverse settings, as detailed below.[19 21 35] Furthermore, the cross-sectional study design precluded the assessment of causal associations. Previous research has identified longitudinal associations of job demands

and resources with burnout,[36 37] which should be clarified for patient-related burnout in particular.

## Comparison with other studies

Our results align with previous findings that job demands are associated with patient-related burnout in general.[14 19] We additionally showed that levels of patient-related burnout were lower among physicians who experienced positive relationships with patients. Physicians perceived positive relationships with patients when they, for example, perceived gratitude from their patients, or felt a strong personal connection with patients. These resources of physicians' work showed to keep physicians going, even in the face of excessive demands.[38] Indeed, physicians experiencing highly positive relationships with patients did not report exhaustion (ie, patient-related burnout), even when exposed to excessive bureaucracy. On the other hand, physicians reporting less positive relationships with patients reported higher levels of patient-related burnout in the face of excessive bureaucracy. In other words, bureaucratic demands are less likely to be associated with patient-related burnout when physicians experience positive relationships with patients; this may indicate positive relationships to buffer the potentially negative impact of excessive bureaucracy on patient-related burnout.

These findings align with those of other research based on the job demands and resources model, which showed that particular resources buffer the negative impact of demands on well-being.[20] Specifically, previous research showed life satisfaction or work engagement are less likely to be impaired by job demands when job resource levels are high.[31 38] In the case of patient-related burnout, the negative impact of bureaucratic demands in particular may be buffered by positive relationships with patients. Bureaucratic demands represent the extent to which physicians perceive policies and procedures as burdensome, ineffective or unnecessary.[26] Bureaucratic demands and the related concept of administrative burden have been shown to be associated with work-related burnout.[10]

Work-related burnout is also more likely in the presence of high workloads,[39–41] which aligns with our study findings showing associations between workloads and patient-related burnout. Alarmingly, these findings may indicate that workloads may exhaust physicians in their core task—caring for patients. Patient-related burnout may, on the other hand, be less likely in the presence of ample development opportunities, consistent with related research[19]; development opportunities have also been shown to benefit physicians' work engagement, another indicator of well-being.[42] Development opportunities stimulate physicians' senses of capability, mastery and skill, which may enhance their sense of clinical competence and prevent stress or exhaustion in the face of clinically or emotionally demanding situations in patient care.[43] Indeed, learning and professional updating in the context of continuous medical education (CME) have been associated with lower levels of stress and burnout.[43]

Therefore, CME activities may be considered when aiming to address patient-related burnout in medical practice.

In this study, patient-related burnout was not affected by job resources involving participation in decision making, leaders' inspiration or supportive collegial relationships. This is consistent with previous research confirming the absence of associations between these resources and physicians' well-being (ie, work engagement and work-related burnout).[42 44 45] The lack of association with collegial relationships is surprising, as collegial and peer support has shown the potential to help physicians deal with stress and threats to well-being.[38] Such support may not be cultivated fully in the professional medical context due to time limitations or physicians' personal barriers (eg, apprehension about peers' views on their ability to cope).[46 47] Full cultivation of collegial support could contribute to a sense of reduced professional isolation and the reduction of physicians' exhaustion in providing patient care.[38 48] For example, collegial support could be fostered by debrief groups in which peers exchange stressful experiences and together reflect on coping with challenges of patient care, for example, emotionally demanding patients.[49]

## Implications for practice and research

Although physicians are generally at low risk of patient-related burnout, even low levels of such burnout may seriously threaten the sustainability of their practice—as manifested by low levels of commitment to work and job satisfaction.[14] Thus, each patient-related burnout risk factor should be prevented or resolved, which could be facilitated by optimising the balance between job demands and resources. This could be achieved by implementing managerial interventions such as worker health surveillance[50 51] or by adapting organisational structures that facilitate job crafting, that is, proactive strategies in increasing job resources and decreasing hindering demands, both at the individual and team level.[52] Hospitals could furthermore consider how to reduce demands that interfere with physician–patient relationships, or invest in professional development programmes (eg, CME) that facilitate physicians' learning and development of positive relationships with patients. Such relationship development could also be fostered by addressing system-level and organisation-level barriers to physicians' delivery of compassionate care (eg, inadequate time with patients, unsupportive leadership, inadequate support personnel and non-facilitative practice structures).[53] Furthermore, positive relationships with patients could be fostered by mindfulness-based communication programmes; these programmes enhanced physicians' dedicated and non-judgmental attention towards patients' thoughts and emotions, which showed to facilitate empathy towards patients.[42 54 55] Such programmes have also been shown to promote physicians' self-compassion and self-care in stressful practice environments, and may thus contribute to physicians' well-being and the prevention of (patient-related) burnout.[42]

Efforts to prevent patient-related burnout should include the creation of a healthy workplace with consideration of

physicians' and patients' input, characterised (according to the work life model) by reasonable workloads, control over the practice environment, stimulating rewards, a supportive community, fair treatment of staff and professional values.[6 56] Care delivery according to professional values may be complicated by physician burnout, as burned-out physicians are more likely to exhibit low levels of professionalism (ie, low adherence to treatment guidelines, lack of professional integrity and low levels of empathy).[57] However, the effects of patient-related burnout on physicians' professionalism remain unclear. Personal-related burnout, that is, the degree to which physicians' exhaustion in their personal life, is associated with physicians' patient-centred attitudes,[58] yet this association has not been studied for patient-related burnout specifically. Furthermore, burnout in general has been associated with suboptimal patient care quality (eg, patient safety and satisfaction)[1 2] yet, future research should also clarify this for patient-related burnout in particular.

**Contributors** RS conceptualised and designed the study; acquired and interpreted study data; drafted the article; and gave final approval of the version to be published. MS contributed to the design of the study; analysed and interpreted study data; drafted the article; and gave final approval of the version to be published. JvdB contributed to the design of the study; interpreted study data; critically revised the article; and gave final approval of the version to be published. KL conceptualised and designed the study; acquired and interpreted study data; critically revised the article; and gave final approval of the version to be published. All authors agree to be accountable for all aspects of the study in ensuring that questions related to the accuracy or integrity of any part of the study are appropriately investigated and resolved.

**Funding** The study was funded by the Dutch Ministry of Social Affairs and Employment.

**Disclaimer** This Ministry had no role in the study design; in the collection, analysis, and interpretation of data; in the writing of the report; and in the decision to submit the article for publication. All authors confirm their independence from funders and had full access to all of study data and take responsibility for the integrity of the data and the accuracy of the data analysis is also required.

**Competing interests** None declared.

**Patient consent for publication** Obtained.

**Ethics approval** Ethical approval was waived by the Medical Ethics Committee of the Amsterdam University Medical Centre (ID XT4-118). All participants gave informed consent before taking part.

**Provenance and peer review** Not commissioned; externally peer reviewed.

**Data availability statement** Data may be obtained from a third party and are not publicly available. No additional data available, as data are protected under contract with participating medical centres. Nonetheless, inquiries about potential research collaboration can be directed to Professor Kiki Lombarts (m.j.lombarts@ amsterdamumc.nl).

**ORCID iD**
Renée Scheepers http://orcid.org/0000-0001-5750-3686

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
