## [Reviewer comments · BMJ Open]

ARTICLE DETAILS

TITLE (PROVISIONAL)	The associations between job demands, job resources and patient-related burnout among physicians: results from a multicentre observational study
AUTHORS	Scheepers, Renee; Silkens, Milou; van den Berg, Joost; Lombarts, Kiki

VERSION 1 – REVIEW

REVIEWER	Makara-Studzińska Marta Jagiellonian University in Krakow, Collegium Medicum, Poland
REVIEW RETURNED	08-Apr-2020

GENERAL COMMENTS	Congratulations to the authors of the research project on the results and conclusions obtained, as they can be an inspiration for other scientists.
---

REVIEWER	Paul Brown University of the West Indies, Mona Jamaica
REVIEW RETURNED	15-Apr-2020

GENERAL COMMENTS	General Thank you for your manuscript. This was an interesting and well-designed study. I just had a few comments, outlined below. Abstract The response rate reported (81.2%) in the Abstract appears to differ from that reported in the Article Summary (71.3%). You concluded that “Health care organisations could consider proactive support of positive doctor–patient relationships to reduce the likelihood of doctors’ patient-related burnout”. However, while acknowledging the strong multi-centre design, as this is a single-country study, I suggest it could be argued that this conclusion overstates the findings, as the latter may not translate to other societies. Perhaps the statement could be moderated. Methods Regarding the research team’s discussion about the theoretical relevance of two job resources items with loadings < 0.40, how do you justify retaining them? Is there any reference to support this? Would also be good to see the factor loadings, as it is unclear
---

	whether all variables included, loaded predominantly onto only one factor. Did not see a STROBE checklist. Discussion There seems to be little discussion on the relationship between workload specifically and burnout.
--	--

REVIEWER	Ajeet Gajra Cardinal Health, USA Employment: Cardinal health Prior employment: ICON Clinical Research
REVIEW RETURNED	26-May-2020

GENERAL COMMENTS	This is a multicenter study evaluating the association between job demands, resources and patient-related burnout from the Netherlands. The study has utilized the CBI for this assessment. Unfortunately, this manuscript does not make for an easy read- it is somewhat chaotic causing the reader to go back and forth repeatedly to try and glean the objectives, methods and outcomes. Minor modifications: Replace the word “doctor” by physician throughout the manuscript. The manuscript could benefit from an English language review- while its well written, the sentence construction can lead to some difficulties in interpretation. E.g. Opening sentence reads: “In the last decade, research has revealed risks of doctor burnout to patient care quality”. I suspect the intent is: In the last decade, research has revealed that physician burnout can negatively impact the quality of patient care rendered. There are several such examples and odd phrase choices e.g. “bureaucratic loads” The sentence “Relationships with patients moderated the association...” Pg2 line 38 and pg 10, line 55 is unclear. Is the intent to state that there was correlation or association etc? Please clarify. Other modifications recommended: The objective is never clearly stated in the text. How is “exhaustion in providing patient care” different from emotional exhaustion and depersonalization? It is not accurate to state that “only two studies have been conducted to date”. As the authors are aware, numerous studies have addressed burnout in physicians. Consider adding references in the background and discussion addressing the work by other groups. The below are some examples- there are several more that can/ should be cited  • Ahmad W, Ashraf H, Talat A, et al. Association of burnout with doctor-patient relationship and common stressors among postgraduate trainees and house officers in Lahore-a cross-sectional study. PeerJ. 2018;6:e5519. Published 2018 Sep 10. doi:10.7717/peerj.5519 • Fagerlind Ståhl AC, Ståhl C, Smith P. Longitudinal association between psychological demands and burnout for employees experiencing a high versus a low degree of job resources. BMC Public Health. 2018;18(1):915. Published 2018 Jul 25. doi:10.1186/s12889-018-5778-x • Messias E, Gathright MM, Freeman ES, et al. Differences in burnout prevalence between clinical professionals and biomedical
--

	scientists in an academic medical centre: a cross-sectional survey. BMJ Open. 2019;9(2):e023506. Published 2019 Feb 19. doi:10.1136/bmjopen-2018-023506 Was only the client related section of the CBI used? Were the Personal and Work-related sections not administered? If not, then why not? If yes, then what were the results? Those results are critical to this investigation and separating out the client-related section is not appropriate. Have those been published elsewhere and is this manuscript an attempt at segmenting the data? Is this a “convenience sample”? How was the sample size determined? If there was no statistical basis for choosing the sample size then that is a major limitation. The population is too heterogeneous: What is the purpose of including a small subset of residents? Their number does not appear sufficient to draw any conclusions for that subset but may well dilute the results of non-residents; consider analysis excluding residents. Similarly for those with “part-time” contract- what was their Full-time equivalent? The views and perceptions of part-time physicians may differ significantly from those working full time when it comes to burnout. This is a limitation. What methodology was utilized to develop the Job resources and job demands sections- would you call that a Delphi process? That these questionnaires were derived from other instruments but themselves are not validated needs to be highlighted as a limitation. Why was CBI chosen over Maslasch? Please provide the rationale and reasoning. In my view, there are significant limitations to this study but if authors can comply with the above then it can be reconsidered, albeit as a new submission.
--	---

REVIEWER	Dr Abdul Samad Dahri Muhammad Ali Jinnah University Pakistan
REVIEW RETURNED	27-May-2020

GENERAL COMMENTS	Overall work is sufficiently elaborated, still, practicality needs enhancement for attracting managerial interventions.
---

VERSION 1 – AUTHOR RESPONSE

#	Reviewers' comments	Response	Location of revisions
	Reviewer #1		
1	Congratulations to the authors of the research project on the results and conclusions obtained, as they can be an inspiration for other scientists.	We thank the reviewer for the positive evaluation of our research project.	Not applicable
	Reviewer #2		
2	Thank you for your manuscript. This was an interesting and well-designed study. I just had a	We are happy to read that the reviewer notes our study to be interesting and well-designed. We	Not applicable

	few comments, outlined below.	thank the reviewer for the feedback and addressed the suggested improvements below.	
3	The response rate reported (81.2%) in the Abstract appears to differ from that reported in the Article Summary (71.6%).	Thank you for noting this, we corrected the response rate in the abstract to 71.6%.	Abstract, page 2
4	You concluded that “Health care organisations could consider proactive support of positive doctor–patient relationships to reduce the likelihood of doctors’ patient-related burnout”. However, while acknowledging the strong multi-centre design, as this is a single-country study, I suggest it could be argued that this conclusion overstates the findings, as the latter may not translate to other societies. Perhaps the statement could be moderated.	We agree with the reviewer that this conclusion may not be generalizable to other health care systems. Therefore, we moderated this statement in the conclusion of the abstract by rephrasing the conclusion in the abstract as follows: “Therefore, positive physician–patient relationships may be supported to reduce the likelihood of physicians’ patient-related burnout. However, the specific support needed to effectively reduce patient-related burnout may vary per healthcare context and thus requires intensified research across health care systems and settings.” We furthermore also reflected on generalizability of the findings in the limitation section of the Discussion. Specifically, we noted that our findings may not be generalizable to other health care settings or systems. Nonetheless our findings did align with related research – on associations between job demands, resources and burnout – in diverse health care settings. ¹⁻⁴ Whether this alignment also applies to patient-related burnout requires intensified research – as we described in the Discussion.	Abstract, page 2 Discussion, page 12
5	Regarding the research team’s discussion about the theoretical relevance of two job resources items	All factor loadings above 0.4 were automatically retained, whereas items with a factor loading below	Methods, page 9 and supplementary

	with loadings < 0.40, how do you justify retaining them? Is there any reference to support this? Would also be good to see the factor loadings, as it is unclear whether all variables included, loaded predominantly onto only one factor.	this predefined threshold were discussed by the research team. We discussed the theoretical importance and the meaning of the items in light of the rest of the instruments used. The items originated from previously validated questionnaires^{5 6} and the research team considered the content of the questions to be relevant for the topic under study. Psychometric follow-up analyses were performed to check the impact of removing the items on the reliability of the used scales and the reliability did not or only marginally improve; therefore, we preferred to maintain the original structure of the validated questionnaire, and we decided to keep the items. We added all factor loadings in supplementary table A1.	table A1
6	Did not see a STROBE checklist.	We included a STROBE checklist as a research checklist file in the resubmission.	Not applicable
7	There seems to be little discussion on the relationship between workload specifically and burnout.	We agree with the reviewer and therefore we added text on the relationship between workload and burnout in the Discussion. We specifically found that our findings on workload and patient-related burnout aligned with previous research on workloads and burnout.⁷⁻⁹	Discussion, page 14
	Reviewer #3		
8	This is a multi-centre study evaluating the association between job demands, resources and patient-related burnout from the Netherlands. The study has utilized the CBI for this assessment. Unfortunately, this manuscript does not make for an easy read- it is somewhat chaotic causing the reader to go back and forth repeatedly to try and glean the objectives, methods and outcomes.	Many thanks for your careful feedback and for providing us with suggestions that help to improve the manuscript. We improved the readability of the manuscript and clarified the alignment between objectives, methods and outcomes by adopting the following changes. First, we explicitly stated our objective in	Entire manuscript

		the Introduction. Then we clarified how this objective was investigated in the Method section, i.e. by conducting a nationwide programme involving measurement of perceived working conditions and well-being of medical staff, specifically by conducting a survey including validated questionnaires about job demands, job resources and patient-related burnout. We finally discussed the main outcomes in relation to the objective of the study at the start of the Discussion.	
9	Replace the word “doctor” by physician throughout the manuscript	We replaced the word “doctor” by the word “physician” in the entire manuscript.	Entire manuscript
10	The manuscript could benefit from an English language review- while its well written, the sentence construction can lead to some difficulties in interpretation. E.g. Opening sentence reads: “In the last decade, research has revealed risks of doctor burnout to patient care quality”. I suspect the intent is: In the last decade, research has revealed that physician burnout can negatively impact the quality of patient care rendered.	We thank the reviewer for noting that the manuscript is well-written and we critically revised the sentence construction in the entire manuscript. We also rephrased the opening sentence in line with the reviewer’s suggestion and rephrased several sentences throughout the manuscript in order to support clear interpretations.	
11	There are several such examples and odd phrase choices e.g. “bureaucratic loads”	We revised unclear phrases in the entire manuscript – revised sentences are marked with track changes – and changed ‘bureaucratic loads’ into ‘bureaucratic demands’.	Entire manuscript
12	The sentence “Relationships with patients moderated the association....” Pg2 line 38 and pg 10, line 55 is unclear. Is the intent to state that there was correlation or association etc? Please clarify.	In this sentence the verb ‘moderated’ refers to a moderation (i.e. interaction) effect¹⁰ of the job resource ‘relationships with patients’ on the association between bureaucratic demands and patient-related burnout. We clarified this	Methods, page 10

		moderation/interaction effect by including the following changes. First, we more extensively explained how we specifically inspected the moderation effect in the Statistical analyses section: “Specifically, we conducted multiple regression analyses including the independent variables (i) a specific job demand (workload or bureaucratic demands), (ii) a specific job resource (participation in decision making, development opportunities, leaders’ inspiration, relationships with colleagues or relationships with patients) and (iii) the interaction term of the respective job demand and resource, and (iv) patient-related burnout as the dependent variable. A significant interaction term indicated a moderation effect, which was inspected by performing simple slopes analysis to measure the conditional effects of the independent variable on the dependent variable for three values of the moderator: (1) low score (-1SD), (2) the average score, and (3) high score (+1SD).” Then we added the following explanation of the moderation effect in the Results section: “Bureaucratic demands were significantly positively associated with patient-related burnout when physicians’ reported low (-1SD; mean of 3.25) or average (mean of 4.00) ratings on the quality of their relationships with patients. Specifically, the association between bureaucratic demands and	Results, page 11 Discussion, page 13
--	--	--	--

		patient-related burnout was stronger when physicians' ratings of relationships with patients were low ($b_{low} = 0.21$; 95% CI, 0.11 to 0.31; $t = 4.18$; $p < 0.001$) than when ratings were average ($b_{average} = 0.10$; 95% CI, 0.02 to 0.18; $t = 2.35$; $p = 0.02$). When physicians' ratings of patient relationships were high, there was no association between bureaucratic demands and patient-related burnout ($b_{high} = 0.02$; 95% CI, -0.09 to 0.13; $t = 0.34$; $p = 0.73$). We also explained this moderation effect by rephrasing text in the Discussion as follows: "Indeed, physicians experiencing highly positive relationships with patients did not report exhaustion (i.e. patient-related burnout), even when exposed to excessive bureaucracy. On the other hand, physicians reporting less positive relationships with patients reported higher levels of patient-related burnout in the face of excessive bureaucracy. In other words, bureaucratic demands are less likely to be associated with patient-related burnout when physicians experience positive relationships with patients; this may indicate positive relationships to buffer the potentially negative impact of excessive bureaucracy on patient-related burnout."	
13	The objective is never clearly stated in the text.	We rephrased the last sentence of the Introduction in order to clarify our objective. Specifically, we noted: "the objective of the current study was to investigate associations of job demands and resources to	Introduction, page 7

		doctors' patient-related burnout, and to clarify the interaction of demands and resources in this context.”	
14	How is “exhaustion in providing patient care” different from emotional exhaustion and depersonalization?	We understand this question of the reviewer; Maslach defined burnout by the dimensions of emotional exhaustion, depersonalization and personal accomplishment. ¹¹ In this study, we focused on a different, yet related conceptualization of burnout, i.e. patient-related burnout, defined by Kristensen as: “The degree of exhaustion that is perceived by the person as related to his/her work with patients.” ¹² Kristensen introduced this conceptualization of burnout in order to provide insight into the degree to which professionals experience exhaustion – considered as a key domain of burnout – in their work with patients. The questionnaire measuring patient-related burnout (the Copenhagen Burnout Inventory) has been validated in diverse professional settings. ¹²⁻¹⁷ In response to the reviewer’s question, we more extensively explained the concept of patient-related burnout in the introduction by discriminating between professionals’ exhaustion in the domain of patient care (i.e. patient-related burnout) from professionals’ exhaustion in their work in general.	Introduction, page 6
15	It is not accurate state that “only two studies have been conducted to date”. As the authors are aware, numerous studies have addressed burnout in physicians. Consider adding references in the background and discussion addressing	We agree with the reviewer that numerous studies have addressed burnout in physicians. In this study we were specifically interested in the concept of patient-related burnout, as research indicated that heavy job	Introduction, page 6 Discussion, pages 13-16

	the work by other groups. The below are some examples- there are several more that can/ should be cited  • Ahmad W, Ashraf H, Talat A, et al. Association of burnout with doctor-patient relationship and common stressors among postgraduate trainees and house officers in Lahore-a cross-sectional study. PeerJ. 2018;6:e5519. Published 2018 Sep 10. doi:10.7717/peerj.5519 • Fagerlind Ståhl AC, Ståhl C, Smith P. Longitudinal association between psychological demands and burnout for employees experiencing a high versus a low degree of job resources. BMC Public Health. 2018;18(1):915. Published 2018 Jul 25. doi:10.1186/s12889-018-5778-x • Messias E, Gathright MM, Freeman ES, et al. Differences in burnout prevalence between clinical professionals and biomedical scientists in an academic medical centre: a cross-sectional survey. BMJ Open. 2019;9(2):e023506. Published 2019 Feb 19. doi:10.1136/bmjopen-2018-023506 	demands and a lack of job resources in medical practice may hinder physicians' time and attention for patient care – the very essence of being a physician. We wondered how this situation might affect patient-related burnout and we found few studies that specifically addressed associations between job demands, resources and patient-related burnout among physicians (see Introduction). We acknowledge the relevance of related studies on burnout in general, and we incorporated studies on burnout in the Introduction and Discussion. Additionally, we included the references suggested by the reviewer in our manuscript (in the Introduction and Discussion) as they were relevant for our synthesis of the literature. We thank the reviewer for these suggestions.	
16	Was only the client related section of the CBI used? Were the Personal and Work-related sections not administered? If not, then why not? If yes, then what were the results? Those results are critical to this investigation and separating out the client-related section is not appropriate. Have those been published elsewhere and is this manuscript an attempt at segmenting the data?	Yes, we only used the client-related section of the CBI, as patient-related burnout was our outcome of interest: our objective was to investigate associations between job demands, job resources and patient-related burnout specifically. Therefore, we did not include the domains work-related burnout or personal burnout of the CBI; they were not outcomes of interest in this study.	Not applicable

		We also prevented the survey being too long or time consuming for physicians as this could have negatively affected response rates, and therefore we did not include domains of the CBI that were not outcomes of interest. We did not publish the results elsewhere and did not segment the data.	
17	Is this a “convenience sample”? How was the sample size determined? If there was no statistical basis for choosing the sample size then that is a major limitation.	Convenience sampling is an appropriate term for our sampling method: all hospitals departments connected to a nationwide online platform could voluntarily participate in our study.¹⁸ Our nationwide sampling approach resulted in a sample covering 50 departments and 14 hospitals in the Netherlands. A sample size analysis (effect size = 0.10, power = 0.80, number of predictors = 11, probability level = 0.5) showed us in advance that a sample of 178 physicians would be sufficient. However, our objective was to also perform multilevel analyses and inspect moderation effects, which require larger sample sizes.^{19 20} Furthermore, we also targeted a larger sample size (resulting in sample of 465 physicians) in order to enhance representativeness of the sample. This study was part of a nationwide program in which participation was voluntary. We considered it undesirable to either deny access to this program at all or to arbitrarily exclude data because of statistical considerations regarding power and sample size. We acknowledge that we cannot exclude selection bias and described this as a limitation of the study.	Discussion, page 12

18	The population is too heterogeneous: What is the purpose of including a small subset of residents? Their number does not appear sufficient to draw any conclusions for that subset but may well dilute the results of non-residents; consider analysis excluding residents. Similarly for those with “part-time” contract- what was their Full-time equivalent? The views and perceptions of part-time physicians may differ significantly from those working full time when it comes to burnout. This is a limitation.	We agree that this is a heterogeneous population; we chose this approach in order to enable a representative sample including physicians with different years of experience, parttime/fulltime contracts, or training levels (including both specialists and residents). We agree with the reviewer that associations between job demands/resources and patient-related burnout may differ for these different subgroups, i.e. residents versus specialists or parttime versus fulltime employed physicians. Therefore, we described in the manuscript that we performed our analyses with covariates including subgroup characteristics (sex, post-MD degree years of experience, full-time/part-time, and type of respondent, i.e. medical specialist/resident); this way we adjusted the analyses for potential confounding of these characteristics. Also, we checked whether these physician characteristics moderated the associations between job demands/resources and patient-related burnout, i.e. whether these associations differed based on physician characteristics. We did not find moderation effects of physician characteristics. Indeed, related research has shown that the association between demands/resources and burnout has been found in heterogeneous professional groups.^{1 2 21-23} We acknowledge that future research could specifically clarify the role of physician characteristics, ranging from training level to specialty; we	Discussion, page 12
----	--	--	----------------------------

		therefore included this as a recommendation for future research in the Discussion.	
19	What methodology was utilized to develop the Job resources and job demands sections- would you call that a Delphi process? That these questionnaires were derived from other instruments but themselves are not validated needs to be highlighted as a limitation.	This methodology to compose the job demands and resources survey was based on mixed (qualitative and quantitative) methods, i.e. a combination of focus groups and a web-based survey. The focus groups provided the possibility to qualitatively explore job demands and resources relevant to medical practice, and the survey enabled quantitative insights (based on rankings) into the job demands and resources that physicians most preferred in the ultimate survey for the study. The survey included validated questionnaires, i.e. the Questionnaire on the Experience and Evaluation of Work, Three-Item Red Tape Scale, the Physician Worklife Survey and the Copenhagen Burnout Inventory.^{5 6} ^{12 24} These questionnaires are composed of different scales (e.g. workload), which have shown valid and reliable as well.^{5 6 12} ²⁴ We also checked the psychometric properties of the scales in our study and found that all scales showed satisfactory to good internal consistency, and inter-scale as well as item-to-total correlations (supplementary tables A1 and A2). We did reflect on the limitations of selecting specific questionnaires on job demands and resources in the Discussion; we specifically noted	Supplementary tables A1 and A2

		that other job demands and resources also may be relevant, especially in other settings (e.g. non-Dutch systems and primary care).	
20	Why was CBI chosen over Maslach? Please provide the rationale and reasoning.	We chose the CBI as it was our objective to study associations between job demands, job resources and patient-related burnout. The domain of patient-related burnout is measured by the CBI. While the Maslach Burnout Inventory includes three components of burnout (emotional exhaustion, depersonalisation, and reduced personal accomplishment); the CBI has its focus on exhaustion and the degree to which exhaustion is attributed to personal life, work in general or working with patients specifically.¹² The latter domain includes the concept of patient-related burnout, which is for example assessed using the item “Are you tired of working with patients?”. We selected patient-related burnout specifically as extensive research had already shown that general burnout (such as measured by the Maslach Burnout Inventory) is affected by job demands and resources, while few studies investigated this topic for patient-related burnout specifically. Patient-related burnout can be problematic for the medical profession as it may undermine physicians’ sense of meaning in the essence of their work: caring for patients.	Not applicable
	Reviewer #4		
21	Overall work is sufficiently elaborated, still, practicality needs enhancement for	We would like to thank the reviewer for the feedback. We	Discussion, page 15

	attracting managerial interventions.	agree that it is valuable to elaborate on managerial interventions in medical practice. We therefore included recommendations about specific managerial interventions that may help to optimize the balance between job demands and resources in relation to patient-related burnout in the Discussion.	
22	[Abstract] Needs no figures. Any specific suggestion/ direction	We agree that the abstract needs no figures, and that a specific suggestion/direction would clarify the conclusion. Therefore, the following suggestion was added to the abstract: “Therefore, positive physician–patient relationships may be supported to reduce the likelihood of physicians’ patient-related burnout. However, the nature and organisation of this support needed to effectively reduce patient-related burnout may vary per healthcare context and thus requires intensified research in diverse health care systems and settings.”	Abstract, page 2
23	[Discussion; comparison with other studies] Add related information in the introduction or hypothesis development section instead of separate section	We agree with the reviewer that findings of related studies should also be included in the Introduction; we indeed included findings of related research (on job demands, job resources and burnout) in the Introduction. These related studies have been pivotal in the development of our objective, i.e. to investigate associations of job demands and resources to physicians’ patient-related burnout, and to clarify the interaction of demands and resources in this context. We also	Not applicable

		reflected on findings of these studies in a separate section in the Discussion, also in line with the journal criteria that prescribe a reflection on the findings in the context of previous findings of related studies.	
24	[References] Add update references	We added several references of recent studies on the topic of job demands, job resources and patient-related burnout.	Entire manuscript

We hope we have adequately answered the questions raised, and sufficiently clarified the changes made in this paper.

We thank you for the opportunity of improving this paper.

Yours sincerely,

Renée Scheepers, PhD

References

1. Bakker AB, Lieke L, Prins JT, et al. Applying the job demands–resources model to the work–home interface: A study among medical residents and their partners. *J Voc Beh* 2011;79(1):170-80.
2. Laschinger HKS, Grau AL, Finegan J, et al. Predictors of new graduate nurses' workplace well-being: Testing the job demands–resources model. 2012;37(2):175-86.
3. Hakanen JJ, Bakker AB, Demerouti E. How dentists cope with their job demands and stay engaged: the moderating role of job resources. *Eur J Oral Sc* 2005;113(6):479-87. doi: 10.1111/j.1600-0722.2005.00250.x
4. Verweij H, van der Heijden FMMA, van Hooff MLM, et al. The contribution of work characteristics, home characteristics and gender to burnout in medical residents. *Adv Health Sc Educ* 2017;22(4):803-18. doi: 10.1007/s10459-016-9710-9
5. Williams ES, Konrad TR, Linzer M, et al. Refining the measurement of physician job satisfaction: results from the Physician Worklife Survey. *Med Care* 1999;1140-54.
6. Veldhoven Mv, Jonge Jd, Broersen S, et al. Specific relationships between psychosocial job conditions and job-related stress: A three-level analytic approach. *Work & Stress* 2002;16(3):207-28.
7. Pastores SM, Kvetan V, Coopersmith CM, et al. Workforce, workload, and burnout among intensivists and advanced practice providers: a narrative review. *Crit Care Med* 2019;47(4):550-57.
8. Bragard I, Dupuis G, Fleet RJEJoEM. Quality of work life, burnout, and stress in emergency department physicians: a qualitative review. *Eur J Emerg Med* 2015;22(4):227-34.
9. West CP, Dyrbye LN, Shanafelt TDJJoim. Physician burnout: contributors, consequences and solutions. *J Int Med* 2018;283(6):516-29.
10. Field A. Discovering statistics using IBM SPSS statistics. London: Sage Publications limited 2013: 395-408.
11. Maslach C, Schaufeli WB, Leiter MP. Job burnout. *Ann Rev Psych* 2001;52(1):397-422.
12. Kristensen TS, Borritz M, Villadsen E, et al. The Copenhagen Burnout Inventory: A new tool for the assessment of burnout. *Work & Stress* 2005;19(3):192-207.

13. Milfont TL, Denny S, Ameratunga S, et al. Burnout and wellbeing: testing the Copenhagen Burnout Inventory in New Zealand teachers. *Soc Ind Res* 2008;89(1):169-77.
14. Fiorilli C, De Stasio S, Benevene P, et al. Copenhagen Burnout Inventory (CBI): A validation study in an Italian teacher group. *Meth Appl Res* 2015;22(4)
15. Molinero ER, Basart HG-Q, Moncada SLJ. Validation of the Copenhagen Burnout Inventory to assess professional burnout in Spain. *Rev Esp Dal Publ* 2013;87(2):165-79.
16. Fong TC, Ho RT, Ng SJTJop. Psychometric properties of the copenhagen burnout inventory—Chinese Version. *J Psych* 2014;148(3):255-66.
17. Mahmoudi S, Atashzadeh-Shoorideh F, Rassouli M, et al. Translation and psychometric properties of the Copenhagen Burnout Inventory in Iranian nurses. *Ir J Nurs Midwif Res* 2017;22(2):117.
18. Professional Performance Online. Retrieved at <https://professionalperformance-amsterdam.com/en/evaluations/tools/>. Accessed 18-6-2020.
19. Shieh GJB. Sample size determination for confidence intervals of interaction effects in moderated multiple regression with continuous predictor and moderator variables. *Beh Res Meth* 2010;42(3):824-35.
20. Maas CJ, Hox JJM. Sufficient sample sizes for multilevel modeling. *Methodology* 2005;1(3):86-92.
21. Bakker AB, Demerouti E. The job demands-resources model: State of the art. *J Manag Psych* 2007;22(3):309-28.
22. Bakker AB, Hakanen JJ, Demerouti E, et al. Job resources boost work engagement, particularly when job demands are high. *J Educ Psych* 2007;99(2):274.
23. Hakanen JJ, Schaufeli WB, Ahola K. The Job Demands-Resources model: A three-year cross-lagged study of burnout, depression, commitment, and work engagement. *Work & Stress* 2008;22(3):224-41. doi: 10.1080/02678370802379432
24. Borry EL. A new measure of red tape: Introducing the three-item red tape (TIRT) scale. *In Publ Manag J* 2016;19(4):573-93.

VERSION 2 – REVIEW

REVIEWER	Paul Brown University of the West Indies, Mona Jamaica
REVIEW RETURNED	05-Jul-202
GENERAL COMMENTS	
Revisions appear adequate.	
REVIEWER	Ajeet Gajra MD FACP Cardinal Health, USA Employment: Cardinal Health
REVIEW RETURNED	14-Jul-2020
GENERAL COMMENTS	
I appreciate the authors painstakingly addressing each reviewer's comments. This has enhanced the quality of the manuscript significantly.